# How Does Health Literacy Modify Indicators of Health Behaviour and of Health? A Longitudinal Study with Trainees in North Germany [note 1]

**DOI:** 10.3390/healthcare10010002

**Published:** 2021-12-21

**Authors:** Peter Koch, Zita Schillmöller, Albert Nienhaus

**Affiliations:** 1Competence Center for Epidemiology and Health Services Research for Healthcare Professionals (CVcare), Institute for Health Services Research in Dermatology and Nursing (IVDP), University Medical Center Hamburg-Eppendorf (UKE), 20246 Hamburg, Germany; Albert.Nienhaus@bgw-online.de; 2Faculty of Life Sciences, Hamburg University of Applied Sciences (HAW), 21033 Hamburg, Germany; zita.schillmoeller@haw-hamburg.de; 3Department for Occupational Medicine, Hazardous Substances and Health Sciences (AGG), German Social Accident Insurance for the Health and Welfare Services (BGW), 22089 Hamburg, Germany

**Keywords:** health literacy, health behaviour, state of health, vocational education, trainees

## Abstract

Background: Health literacy (HL) is a resource that can help individuals to achieve more control over their health and over factors that influence health. In the present follow-up study, we have investigated the extent to which HL in trainees changes over time and whether or to what extent HL influences health behaviour and health. Methods: In 2017, we performed a baseline survey (T0) of trainees from six different branches, who were contacted through vocational colleges in four northern federal states in Germany. The survey was repeated at the midpoint of their training in 2019 (T1). Demographic data were surveyed, together with information on HL (HLS-EU-Q16), health behaviour and on health status (psychological well-being, subjective health status). Multivariate regression analyses were performed in SPSS 26. Results: Three hundred and ninety-one (391) data sets were evaluated, with a follow-up rate of 27%; 79% of the trainees were female. The mean age was 21.2 years. Over all subjects, the mean HL increased over time ($\bar{x}$ (SD): 11.9 (2.9) to 12.2 (2.9), *p* = 0.070). This increase was only statistically significant for the health service trainees ($\bar{x}$ (SD): 12.1 (2.8) to 12.5 (2.9), *p* = 0.019). Relative to persons with adequate HL, the odds ratio over time for impaired psychological well-being was increased by 230% in persons with inadequate HL (OR: 3.3, 95% CI: 1.70–6.32, *p* < 0.001). For persons with problematical HL, the corresponding increase in odds ratio was 110% (OR: 2.1, 95% CI: 1.30–3.38, *p* = 0.002). Relative to persons with adequate HL, trainees with inadequate HL exhibited a significant increase in odds ratio of 2.8 over time for poor or less good subjective health status (OR: 2.8, 95% CI: 1.23–6.33, *p* = 0.014). Conclusions: We observed a positive longitudinal association between HL and health. A significant increase in HL was observed in trainees in the health service. Thus the study shows that the concept of HL may provide a potential preventive approach for trainees.

## 1. Introduction

In recent years, there have been many studies on the theme of health literacy (HL). HL represents the resources and potentials which allow individuals to assert more control over their health, as well as over factors that influence health [1]. The demand for HL has been incorporated into documents on political strategy, such as the European Health Concept 2020 [2] or the National Action Plan for Health Literacy in Germany [3].

### 1.1. Definition of Health Literacy (HL)

Many models for HL have been published. In their review of the literature, Sørensen et al. developed a definition of HL based on 12 conceptual models and 17 definitions [4], page 3: “Health literacy is linked to literacy and entails people’s knowledge, motivation and competences to access, understand, appraise and apply health information in order to make judgments and take decisions in everyday life concerning healthcare, disease prevention and health promotion to maintain or improve quality of life during the life course.” In this context, HL is regarded as an individual dynamic ability, and can be learnt. It is related to the health system and the needs of this system. Should the external conditions or individual circumstances change, the acquired abilities must be appropriately adapted or extended [5].

### 1.2. Prevalence of Limited HL, Determinants and Associated Outcomes

HL influences both well-being and health [6]. International studies have identified deficits in HL in the populations of many countries. For the European region, the European Health Literacy Survey performed in 2011 found on average about one in two people has limited HL [7]. Of the eight European countries studied, study participants from Bulgaria had the highest proportion of limited HL (62%), and the second highest proportion was observed among study participants from Austria (56%). The lowest proportion of impaired HL was observed in the Netherlands (29%) and Ireland (40%). With a share of 46%, the German participants were in the middle of the European comparison. In more recent studies from Germany, more than half the population (54%) exhibit impaired HL [8]. Indeed, in the current HLS-GER 2 study (2020), this fraction rose to 59% [9]. In particular, socio-economic factors seem to be linked to impaired HL. International studies have found associations with low levels of education, low social status and background or experience of immigration [3,7,9,10,11,12]. In a German sample, Berens et al. found decreasing HL with increasing age [13]. Impaired HL is also associated with a poor health status and poor health behaviour, as well as low participation in health activities and increased demand on health services [6]. Associations between HL and health indicators have been identified in both the general population and in various subgroups, including socioeconomically disadvantaged adults, patients, nursing staff and their children and adolescents [14,15,16,17,18,19]. As health behaviour is apparently a mediator in the relationship between HL and health, there is also published evidence for an association between HL and health behaviour [20,21,22,23]. However, this relationship is clearer in studies of adults than of adolescents [20].

### 1.3. Target Group

Dedicated studies are planned to test the concept that HL can help to stimulate a healthy working environment, once applied to the workplace and educational system [2,3]. There are currently about 1.3 million trainees in Germany in dual vocational training in companies and vocational colleges [24]. Each year, about a quarter of these occupational training contracts are broken off prematurely (see reference above). Many newcomers find that starting vocational training is a challenge. The demands and processes at work present new psychological and physical sources of stress, as well as novel social demands. These young people are at risk, as they are still maturing and generally do not believe that unhealthy behavioural patterns have unfavourable consequences for them, as well as not being generally aware of issues of safety at work and health protection [25]. These new demands may affect the employees‘ health behaviour and health status. Studies have shown that these demands have unfavourable effects on eating habits and the extent of physical activity [26]. Moreover, health problems, such as pain in the back or headache, may be associated with activities at work [27]. With good HL, the employee becomes aware of how their work may affect their health and can help to reduce accidents at work and occupational diseases [3,6].

### 1.4. Research Question

Measures to improve HL must be based on the conditions and needs of the target group [1]. Within the German system of vocational training, HL has only been studied for university graduates [28]. In order to develop methods within vocational training, the first step is to describe the occurrence of HL in trainees. In the published baseline investigation of the present longitudinal study, limited HL was observed in 53% of trainees in six different branches. More details on this study can be found in the corresponding publication [29]. As there has been little research on HL in occupational trainees, we regard this study as an essential contribution to research on health support and prevention in adolescents in Germany. Due to the small number of studies on HL and health outcomes in young adolescents and the inconsistency of these study results [22], we would like to provide further evidence for this research topic. This also concerns the inconsistent study results on HL and health behaviour in adolescents, which are in contrast to the consistent study results in adults [23].

The objective of our study was to examine the following hypotheses:(i)Trainees with high initial HL at T0 exhibit in a longitudinal section better health behaviour at T1 (alcohol consumption, smoking, nutrition, exercise) than trainees with low initial HL.(ii)In a longitudinal section, trainees with high initial HL at T0 exhibit better health at T1 (BMI, subjective health status, psychological well-being) than trainees with low initial HL.

For hypothesis generation, we formulated the following research question: How are the changes of HL, health behaviour and health in the different subgroups (branches, genders) over time?

## 2. Materials and Methods

### 2.1. Study Design and Cohort

The study has a prospective cohort study design. The trainees came from six different branches:Health: geriatric nurses, medical nurses, medical assistants;Cosmetics: hairdressers;Education: educators;Engineering: plant mechanics for plumbing, heating and air conditioning, electricians for plant and building technology;Office: office management assistants, wholesale and export sales managers, industrial sales manager;Retail sales: retail sales manager.

The baseline survey for the present study was performed in 2017. For this purpose, we contacted all vocational colleges in the federal states in North Germany (Schleswig-Holstein, Bremen, Lower Saxony and Mecklenburg-Western Pomerania) that offered relevant courses—as identified by an internet search. Approval was not granted by the educational authorities of the federal state of Hamburg. We obtained an ethics vote from Hamburg Medical Association (PV5670). Further details can be found in the publication on the cross-sectional study [29].

Forty seven of the 321 identified vocational colleges agreed to participate in this study (response rate 14.6%). In the course of October 2017, 5052 trainees were invited to participate in the study; 1797 trainees returned the questionnaire to the study centre (response rate 35.5%). At the midpoint of the training (March 2019), a printed questionnaire form was sent to the private addresses of 1569 trainees who had given their approval for follow-up surveys. A total of 422 questionnaires were returned to the study centre (follow-up rate 27%).

### 2.2. Survey Instrument

The survey covered sociodemographic data on age, gender, country of birth, nationality and school leaving exam.

#### 2.2.1. Health Literacy

HL was determined with a validated short questionnaire, the Health Literacy Survey Questionnaire (HLS-EU-Q16) [30]. Within the areas of coping with illness, prevention and health support, the 16 items cover the four abilities in dealing with health-related information—identification, understanding, assessment and application. The four-step answer categories were dichotomised and a total score of 0 to 16 points was calculated. According to literature HL was classified into 3 levels: adequate (13–16 points), problematic (9–12 points) and inadequate (0–8 points). If values were missing for more than two items, the total score was rated as missing.

#### 2.2.2. Health Status

Four different indicators of health status were surveyed. In addition to the BMI, the subjective health status was estimated using a five-step assessment (excellent/very good/good/less good/poor) [26] and dichotomised as follows good: excellent/very good/good and poor: less good/poor. Various diseases with a medical diagnosis in the preceding 12 months were surveyed using the items in the Work Ability Index [31]. Psychological well-being was surveyed using the WHO 5 index [32]. The WHO 5 Index was dichotomised using the cut-off scale as described in the literature (poor psychological well-being: <13).

#### 2.2.3. Health Behaviour

The frequency of sporting activity was surveyed on the basis of five categories (none/<1 h per week/1–<2 h per week/2–<4 h per week/≥4 h per week) [33]. Smoking behaviour and alcohol consumption were assessed [34,35] A food frequency questionnaire was used to survey nutritional behaviour [36]. A nutritional score was formed based on 15 food groups. For each of the six steps of frequency of consumption (daily to never), between 0 and 2 points (abnormal, normal, optimal frequency of consumption) were awarded for each of the surveyed food groups. A score between 0 and 30 was calculated by adding the points. The following categories were then assigned: optimal nutritional pattern (16–30 points), normal nutritional pattern (13–15 points) and unfavourable nutritional pattern (0–12 points).

The variable that summarised health behaviour was then calculated as follows: in the first step, the five variables related to health behaviour were dichotomised and coded as follows:Sporting activity yes (1): 2–<4 h per week/> = 4 h per week, no (0): none/<1 h per week/1–<2 h per weekNutrition favourable pattern (1): normal/optimal pattern, unfavourable (0): unfavourable patternFast Food: no (1): <= once/week, yes (0): >once/weekSmoking no (1), yes (0)Risky alcohol consumption no (1), yes (0)

In the second step, these variables were then added up to form a total score for health behaviour with the following range: 0 (unfavourable)–5 (favourable).

### 2.3. Statistical Analysis

Groups of nominal data were compared with Pearson’s chi^2^ test. The *t* test was used for normally distributed data. The Mann Whitney U test was used to compare non-normally distributed metric data. In order to examine the time courses of HL, indicators of health and health behaviour, tests for dependent data were calculated—the McNemar test for nominal data, the Wilcoxon test for non-normally distributed data and the *t* test for normally distributed variables.

Hypotheses were tested by multivariate logistic regression. Effect estimates were reported with 95% confidence intervals. The level of significance was *p* < 0.05. The statistical analysis was performed with SPSS 26 (IBM Corp., Armonk, NY, USA).

## 3. Results

### 3.1. Description of the Study Cohort

Of the 422 participating trainees, 27 reported that they had broken off their training, and four persons could not be assigned. Thus 391 persons were included in the analysis. 79% of the trainees were female. At the time of the baseline survey, the mean age was 21.2 years (SD: 5.1) (Table 1). 95% of the trainees reported that they were of German nationality and 5% of another nationality. The most frequent school leaving exam was the higher secondary school (Realschulabschluss) (47%), then A-Levels (Abitur) (29%), then school leaving exam for the vocational training college (Fachhochschulabschluss) (19%) and school leaving exam for the lower secondary school (Hauptschulabschluss) (5%). The group of Nursing/medical assistants was the largest branch—at 42% -, followed by the group of Office (31%). Smaller contributions were made by the groups of Education (12%), Retail trade (8%), Engineering (4%) and Hairdressers (3%). Women were dominant in all training areas, except for engineering, where there were only men. The highest proportion of women was in the group nursing/medical assistants (95%). Most trainees came from the vocational training schools in Lower Saxony (76%) or in Schleswig-Holstein (20%). Only 4% of trainees came from vocational training colleges in Mecklenburg-Western Pomerania and none of the trainees of vocational schools in Bremen took part in the follow-up assessment.

### 3.2. Health Literacy

Table 2 shows the values of HL for the different demographic characteristics at the time of the baseline survey. There was no difference between the genders in HL. Mean values of HL decreased with age. In the youngest age group (16–18 years), mean HL was 12.4. This value then fell continuously with age, reaching 11.4 in the oldest age group (≥26 years). Persons of German nationality had a slightly lower HL than persons of other nationalities (11.9 vs. 12.1). HL tended to increase slightly in trainees with a better school leaving exam. Thus trainees with a lower secondary school leaving exam (Hauptschulabschluss) exhibited a lower mean value than trainees with A-Levels (Abitur) (10.9 vs. 12.2). None of these differences was statistically significant.

Table 3 shows how HL, health behaviour and state of health change over time. Changes in HL: In addition to the total group, changes over time were tested for the subgroups Branch and Gender, although the table only presents statistically significant changes over time. For the mean of the HL score, there was a slight increase over time, but this was not statistically significant (11.9 vs. 12.2, *p* = 0.070). In the categorical form of the variable, it was found that there was a slight increase over time in the proportion of trainees with adequate HL (47% vs. 51%) (Figure 1). In the branches of the health service and welfare work (education, nursing/medical assistants, hairdressers), there was an increase in HL, although this was only statistically significant in the subgroup of nursing/medical assistants—from 12.1 to 12.5 (*p* = 0.019) (Table 4).

### 3.3. Health Behaviour and Health Status

Unfavourable nutrition was found for about 50% at both time points. At both time points, the consumption of fast food was about 15% (Table 3). The prevalence of smoking was about 30% at both the baseline survey and the follow-up. The proportion of those with less than two hours exercise per week fell slightly from 64% to 62%. The proportion of those with risky alcohol consumption also fell (46% vs. 41%), and this was particularly marked for women (44% vs. 37%, *p* = 0.024).

BMI increased significantly for the whole group over time (23.9 vs. 24.3, *p* = 0.033). This was also found for the subgroups hairdressers and office (24.4 vs. 25.9, *p* = 0.014 and 23.5 vs. 24.0, *p* = 0.020, respectively). At both time points, the proportion of those with poor or less good subjective health status was about 15%.

Musculoskeletal diseases (MSD) were present in 21% or 18%, respectively; the proportion of skin diseases fell from 22% to 14% (*p* = 0.002) overall and in the subgroups Office and Women from 25% to 10% (*p* = 0.002) and from 26% to 15% (*p* = 0.011), respectively. No statistically significant changes were found for diseases of the respiratory tract (21% vs. 19%), the psyche (10% vs. 9%), neurological diseases (16% vs. 14%) or diseases of the digestive system (8% vs. 11%). There were significant increases over time in hormonal diseases, both in the whole population (10% vs. 14%, *p* = 0.008), and in the subgroups Nursing/Medical assistants (9% vs. 18%, *p* = 0.001) and Women (11% vs. 16%, *p* = 0.021). There was hardly any change in cardiovascular diseases over time (5% vs. 6%). Intermediate values were found for psychological well-being and these were constant over time (13.4 vs. 13.7).

At the time of the follow-up, there was no significant difference in HL between the groups of men and women ($\bar{x}$ (SD) Men: 12.6 (3.0), Women: 12.1 (3.0), *p* = 0.197; not shown in the table). At the time of the follow-up, the following differences in health behaviour were observed: Men exhibited risky alcohol consumption more often than women (Men: 53%, Women: 37%, *p* = 0.010) and more often smoked than did women (Men: 38%, Women: 28%, *p* = 0.083). Lack of exercise was more often found for women than for men (Men: 44%, Women: 66%, *p* < 0.001). Fast food consumption or unfavourable nutrition were more often observed for men than for women (Men: 23%, Women: 12%, *p* = 0.010 and Men: 55%, Women 47%, *p* = 0.185, respectively).

BMI ≥ 25 was slightly more often found for women than for men (Men: 34%, Women: 37%, *p* = 0.554). Lower psychological well-being was observed for women than for men ($\bar{x}$ (SD) Men: 14.9 (4.5), Women: 13.4 (4.5), *p* = 0.009). At the time of the follow-up, women more often reported poor subjective health status than men (Men: 11%, Women: 16%, *p* = 0.234). As regards medically diagnosed diseases, women significantly more often reported musculoskeletal diseases (Men: 10%, Women 20%, *p* = 0.046) and neurological diseases (Men: 6%, Women: 16%, *p* = 0.022).

### 3.4. Associations of Health and Health Behaviour with HL

Multivariate logistic regression showed that HL was a statistically significant predictor of psychological well-being (Table 5). Thus, trainees with inadequate HL have—in comparison to persons with adequate HL—an increased odds ratio of developing low psychological well-being, with an increase in odds ratio of 230% for inadequate HL (OR: 3.3, 95% CI: 1.70–6.32, *p* < 0.001) and of 110% for problematic HL (OR: 2.1, 95%CI: 1.30–3.38, *p* = 0.002). In comparison to men, the odds ratio for women was increased to 1.9, which was statistically significant (OR: 1.9, 95% CI: 1.06–3.35, *p* = 0.032).

For subjective health status there was a statistically significant increased odds ratio of 2.8 for a poor or less good health status for trainees with inadequate HL, relative to trainees with an adequate health status (OR: 2.8, 95% CI: 1.23–6.33, *p* = 0.014) (Table 6). Persons with problematic HL had an increased odds ratio of 1.6, which was not statistically significant (OR: 1.6, 95% CI: 0.84–3.22, *p* = 0.147). It was also found that persons with moderate health behaviour at T0 (OR: 2.2, 95% CI: 1.11–4.44, *p* = 0.023) or poor health behaviour (OR: 4.0, 95% CI: 1.51–10.48, *p* = 0.005) were also a risk group for unfavourable subjective health status at T1—in comparison to persons with good health behaviour.

As regards BMI, multivariate analysis found an inverse association—so that adequate HL was associated with high BMI. The findings had very wide confidence intervals and are difficult to interpret. These results are not shown. No associations were found for the indicators of health behaviour at T1 (alcohol, smoking, exercise, fast food and nutrition) and HL at T0. Figure 2 shows the missing associations by means of HL (T0) and HL (T1) box plots in classes of the health behaviour total score (T1). Moreover, no associations were observed between the classes of health behaviour at T1 and health literacy at T1.

## 4. Discussion

In summary, this longitudinal study found that inadequate HL at T0 was a significant factor in relation to poor psychological well-being and poor subjective health status at T1. There was no support for the working hypotheses that HL had a significant effect on BMI or health behaviour.

The observed prevalence of limited (problematic or inadequate) HL at T1 was 49%, which was less than the prevalence in the baseline study (53%) [29] and is also lower than some values from the German speaking area. The most recent survey of the adult German population found a value of 59% [9], but Jordan and Hoebel reported a prevalence of 44% in 2015. The prevalence of limited HL has also been reported for younger subjects, namely 58% in 15-year-old Austrians and 69% students of a German college of health [28,30]. It is possible that the students from the college of the health sector answered a little more critically from their own perspective, so that the proportion of limited HL is higher in this case than in comparison with samples from this age group. In a survey conducted in Germany in 2014, Berens et al. found a similarly high prevalence of limited HL (47%) for the age group of 15–29 years as in our study [13]. In another German study performed in 2016, an equally high prevalence of 47% was observed among students aged 20–29 [37]. In summary, the observed prevalence of limited HL appears to be in line with existing study findings.

### 4.1. HL and Demographic Parameters

Even though there have been published reports of associations between HL and demographic characteristics, there were no significant correlations in the present study [7]. An inverse trend was observed for age group, which has a very limited range in our study. The mean HL decreased with increasing age. In contrast, the most recent German population-related study (Schaeffer et al.) recorded a continuous increase in HL within the range from 18 to 64 years, followed by a decrease from 65 years of age [9]. This discrepancy can certainly be explained by the narrow age range in the present study. This finding is probably unsystematic. On the other hand, one cannot exclude that, within this age range, the trainees acquire experience in health and the health system with age. They become increasingly conscious of their own ignorance and this leads to lower subjective HL. As HLS-EU-Q16 is a subjective instrument for measuring HL and is thus a self-evaluation, the possibility remains that some individuals regard their HL as good, due solely to their deficient experience or knowledge—even though their HL is objectively poor. In the study of Schaeffer et al. (see reference above), immigrants exhibited lower values of HL, but we did not observe this in our group. It is possible that the subgroup was too small to map this effect. It must also be said that the comparison of an adult sample with our data is of course limited. There was a favourable trend with level of education, used in our study as a proxy for socioeconomic status. This is in agreement with the published literature, although significant differences were not found in our study.

### 4.2. HL and Health Education

The present study investigated changes in HL over time in six different branches. An improvement in HL was observed in professions related to health education, particularly nursing/medical assistant. It is plausible that professional training related to health education has a more positive effect over time on the development of health literacy than do commercial, technical or administrative training. When looking through the framework curricula of the different training professions in the six branches, we discovered that the plan for nursing professions is the only one that deals with one’s approach to health. Similarly, in a cross-sectional study, Sukys et al. showed that students in health-related university courses exhibited better HL values than students in other courses—based on HLS-EU-Q47 [38]. The same conclusion was reached by the authors of a Danish cross-sectional study, who observed that students in health education also exhibited better HL values than students in other courses [39]. Therefore, we cannot exclude that the observed increase in HL for the group nursing/medical assistants is a consequence of dealing with the health of others and one’s own health in the setting of the vocational school and company. Appropriate studies need to be carried out to verify this indication.

### 4.3. HL and Health Outcomes

Very few studies have examined the association between HL and health in adolescents. The present study found an odds ratio of 3.3 of having poor psychological well-being for persons with inadequate HL and of 2.1 for persons with problematic HL. This association of HL with psychological well-being is confirmed in the study of Björnsen et al. on adolescents [15]. This cross-sectional study was performed on Norwegian pupils aged 15–21 years and found that high mental HL was associated with high psychological well-being. This association has also been confirmed in other cross-sectional studies with adults. Zhang et al. observed this association in a population-related sample from Hong Kong. Analogous findings were reported by Fiedler et al. in German industrial managers and by Amoah et al. in a population-related study in Ghana [19,40,41]. In summary, our findings confirm the few existing studies on the relationship between HL and psychological well-being in adolescents.

The results on the association between HL and subjective health status are evident as a trend in the odds ratios in the three-step HL variable. Trainees with an inadequate HL have a 2.8-fold greater chance of low subjective health than persons with adequate HL. For persons with problematic HL, this chance is raised 1.6-fold, although the differences are not statistically significant. There have hardly been any published studies on the influence of HL on subjective health in this target group. There have been two studies with vocational college students and these show that young adults with high HL also exhibit better subjective health than their fellows with low HL [39,42]. In a systematic review, Sansom-Daly et al. concluded that there have been very few studies on the association between HL and various health indicators in adolescents—and that the study results were not consistent [22]. With regard to the relationship between HL and subjective health, our study adds positive results to the few existing and controversial studies. Furthermore, our study provides results for both questions regarding psychological well-being and subjective health based on longitudinal data.

### 4.4. HL and Health Behaviour

No associations could be found between HL at T0 and the various indicators of health behaviour at T1, including the cumulative score of health behaviour. It is also possible to conceive that HL has immediate effects on health behaviour—which could be investigated by testing HL at T1 relative to health behaviour at T1, but this approach also failed to identify any associations. The lack of association between HL and health behaviour is somewhat inconsistent with the results of previous studies. In a systematic review, Fleary et al. reported that significant associations between HL and health behaviour were observed in 13 of 17 studies on adolescents [20]. As a qualification, it must be mentioned that the definitions, instruments and theoretical foundations of these studies were very different. Thus, the various instruments in these studies surveyed functional and medial HL either subjectively or objectively (through a performance test). In a German study on adolescents with limited HL—as surveyed with a long version of the questionnaire used here (HLS-EU-Q47)—no association was found with consumption of tobacco or alcohol, but with nutrition and exercise [43]. Moreover, in a study with 15-year-old Austrians, no association could be found with HL for two of three health indicators (alcohol consumption and smoking; the association with exercise frequency was only weak (r = 0.14) [30]. On the other hand, a systematic review of interventions in adults found a clear association between HL and the health behaviour outcome [23]. Thus, taken together, the literature gives disparate results for this association. In the present study and because of the age of the subjects, it remains possible that the lack of identifiable association between HL and health behaviour may be due to subjective misassessment of the HL. There is also evidence that the adolescents did not understand all formulations employed in the long version of the questionnaire used (HLS-EU-Q47), even though they answered these questions during the study [44]. In the final analysis, we cannot explain why this association was not found in this study. It might be important to achieve further insights by using both subjective and objective measurement instruments for HL, as was recommended by Okan in his inventory of HL instruments for children and adolescents [45]. It would then be expedient to perform qualitative research to examine the extent to which adolescents with good subjective and objective HL are prepared to accept unhealthy health behaviour.

### 4.5. Prevention

The present study includes findings for trainees in different branches. An increase in BMI was found both in the total group and particularly in the branches of Office and Hairdressing. For women in particular, there was a major lack of exercise and poor psychological well-being. The male trainees tended to risks alcohol consumption, smoking and unhealthy nutrition. Group-specific prevention programmes should be supported to encourage healthy behaviour in vocational colleges and at work. Moreover, there should be interventions to improve HL. This is particularly the case for branches that are unrelated to health. There have already been successful reports of studies to improve HL in the setting of schools [46,47].

### 4.6. Limitations

To the best of our knowledge, this is the first longitudinal study to be published that examined the influence of HL on the indicators of health behaviour and health in trainees.

As there was a relatively high proportion of drop-outs in this study, there is a possibility of selection bias. The drop-out analysis showed that persons who reported at baseline that they had selected their profession in an emergency were likely to have an increased risk of drop-out and therefore did not really belong to the target population. Another factor predicting dropout was male gender, with a lower proportion of males in the follow-up than at baseline (21% vs. 30%). Regarding the health indicators (BMI, psychological well-being, subjective health status), there were no differences between the baseline sample and the follow-up sample, for the health behaviour (diet, smoking, alcohol consumption, exercise), the follow-up sample showed a lower proportion of smokers than the baseline sample at T0 (30% vs. 42%). Moreover, the relatively low response rate in the baseline study showed that the degree of generalisability is limited. For some of the subgroups (hairdressers, mechanics, retail), the case numbers are very low, but the subgroups could not be combined, as these branches are very different. The low variance assignment in the two multivariate models shows that some variables are missing in the model that could help to explain the variance. However, as the study was not designed to be exploratory and several factors were explored that influenced the outcome, we consider that this was only a minor limitation. Independent and dependent variables come from the same source, so that bias from common-method variance is possible. It should also be emphasised that the health information is subjective.

## 5. Conclusions

This study shows that the concept of HL presents a potential preventive approach in trainees, as it is longitudinally associated with health. A positive development of HL was observed in trainees whose training curriculum was related to health. This was not the case for other branches. This shows that HL in trainees is a variable parameter and presents a possible target if appropriate information is imparted. However, this exploratory result should be used in further appropriate studies to examine the extent to which curricula of health-related occupations actually increase HL over time compared to other curricula. This study showed that trainees with inadequate or problematic HL, female trainees, and persons with unfavourable health behaviour are risk groups within this study population, who should be assured better access to health support and prevention. For these at-risk groups, school and workplace-based intervention programmes that also increase HL should be implemented. As no association was established between HL and health behaviour, future studies must also consider objective HL as an additional factor, in order to control for subjective faulty assessment. It would also be desirable to employ a qualitative design to investigate to what extent health-competent young people are deliberately willing to accept unfavourable health behaviour.

## Figures and Tables

**Figure 1 healthcare-10-00002-f001:**
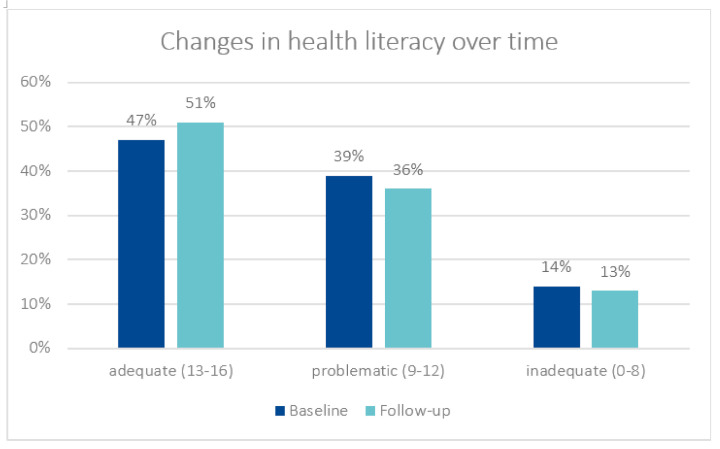
Changes in health literacy over time (*p* = 0.315).

**Figure 2 healthcare-10-00002-f002:**
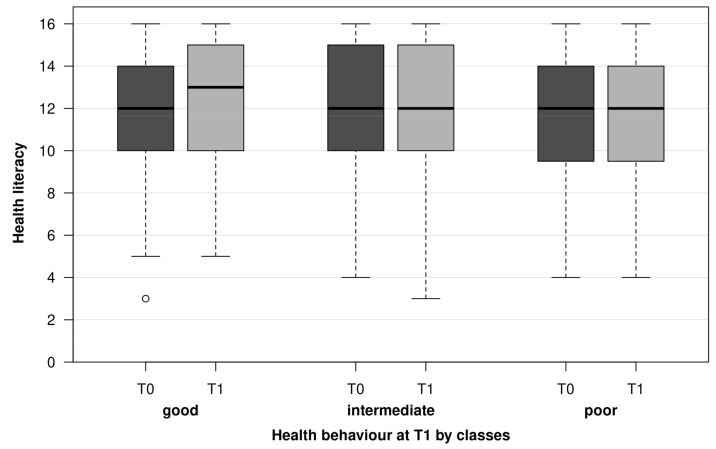
Distribution of HL at T0 and T1 over health behaviour into 3 classes at T1.

**Table 1 healthcare-10-00002-t001:** Demographic characteristics of the trainee cohort.

	Female 309 (79%)	Male 82 (21%)	Total 391 (100%)	*p*
Age in years (baseline)				
$\bar{x}$ (SD)	21.2 (5.4)	21.0 (3.8)	21.2 (5.1)	0.328 ^1^
Range	16–53	13–36	16–53	
Nationality				
German	290 (79%)	76 (21%)	366 (95%)	0.934 ^2^
Other	16 (80%)	4 (20%)	20 (5%)	
*n* missing	3	2	5	
School leaving exam				
Lower secondary school (Hauptschule)	13 (62%)	8 (38%)	21 (5%)	0.069 ^2^
Higher secondary school (Realschule)	150 (82%)	32 (18%)	182 (47%)	
Vocational training college (Fachhochschule)	62 (83%)	13 (17%)	75 (19%)	
A-Levels (Abitur)	84 (74%)	29 (26%)	113 (29%)	
Branch				
Office	76 (63%)	45 (37%)	121 (31%)	<0.001 ^2^
Retail	21 (70%)	9 (30%)	30 (8%)	
Education	43 (92%)	4 (8%)	47 (12%)	
Nursing/Medical assistants	158 (95%)	9 (5%)	167 (42%)	
Engineering	0 (0%)	14 (100%)	14 (4%)	
Hairdressers	11 (95%)	1 (8%)	12 (3%)	
Federal State/Vocational training college				
Lower Saxony	238 (80%)	59 (20%)	297 (76%)	0.275 ^2^
Mecklenburg-Western Pomerania	14 (87%)	2 (13%)	16 (4%)	
Schleswig-Holstein	57 (73%)	21 (27%)	78 (20%)	
Bremen	0 (0%)	0 (0%)	0 (0%)	

^1^ Mann–Whitney U Test. ^2^ Pearson’s Chi^2^ Test.

**Table 2 healthcare-10-00002-t002:** Health literacy (HL) for different demographic variables.

Demographic Variables	Category (*n*)	$\mathrm{HL}\;\mathrm{Score}\;\mathrm{Baseline}\;\mathrm{(SD) (SD)}\;\bar{x}$	*p* *
Gender	female (309)	11.9 (3.0)	0.865
	male (82)	11.9 (3.0)	
Age group (years)	16–18 (102)	12.4 (2.9)	0.093
	19–20 (145)	12.0 (2.9)	
	21–25 (98)	11.5 (3.1)	
	≥26 (44)	11.4 (3.1)	
Nationality	German (366)	11.9 (3.0)	0.790
	Other (20)	12.1 (3.4)	
School Leaving Exam	Lower secondary school (Hauptschule) (21)	10.9 (3.6)	0.304
	Higher secondary school exam (Realschule) (182)	11.9 (3.0)	
	Vocational training college leaving exam (Fachhochschule) (75)	11.8 (3.1)	
	A-Levels (Abitur) (113)	12.2 (2.7)	

* ANOVA.

**Table 3 healthcare-10-00002-t003:** Changes over time in HL, health behaviour and health status.

Variable		Group	Baseline $\bar{x}$ **(SD)** *n* (%)	Follow-Up $\bar{x}$ **(SD)** *n* (%)	*p*	Trend
Health literacy (Score 0–16)		Total	11.9 (2.9)	12,2 (2.9)	0.070 ^2^	
		Nursing/ MFA	12.1 (2.8)	12.5 (2.9)	0.019 ^2^	**↑**
Unfavourable nutrition		Total	188 (49%)	184 (48%)	0.696 ^1^	
Fast food		Total	60 (15%)	56 (14%)	0.708 ^1^	
Smoking		Total	117 (30%)	116 (30%)	0.885 ^1^	
Lack of exercise (<2 h exercise/week)		Total	249 (64%)	240 (62%)	0.542 ^1^	
Risky alcohol consumption		Total	175 (46%)	156 (41%)	0.073 ^1^	
		Women	133 (44%)	113 (37%)	0.024 ^1^	**↓**
BMI		Total	23.9 (5.1)	24,3 (4.9)	0.033 ^1^	**↑**
		Hairdressers	24.4 (5.8)	25.9 (7.0)	0.014 ^1^	**↑**
		Office	23.5 (4.5)	24.0 (4.9)	0.020 ^1^	**↑**
Subjective health status (poor/less good)		Total	59 (15%)	56 (14%)	0.804 ^1^	
Medically diagnosed diseases	Musculoskeletal disease (MSD)	Total	79 (21%)	69 (18%)	1.000 ^1^	
	Skin	Total	84 (22%)	54 (14%)	0.002 ^1^	**↓**
		Office	30 (25%)	12 (10%)	0.002 ^1^	**↓**
		Women	69 (26%)	46 (15%)	0.011 ^1^	**↓**
	Respiratory tract	Total	83 (21%)	74 (19%)	0.306 ^1^	
	Psyche	Total	40 (10%)	34 (9%)	0.361 ^1^	
	Neurological	Total	60 (16%)	55 (14%)	0.519 ^1^	
	Digestive system	Total	32 (8%)	41 (11%)	0.203 ^1^	
	Hormonal	Total	38 (10%)	54 (14%)	0.008 ^1^	**↑**
		Nursing/ Medical Assistant	15 (9%)	30 (18%)	0.001 ^1^	**↑**
		Women	35 (11%)	48 (16%)	0.021 ^1^	**↑**
	Cardiovascular	Total	18 (5%)	22 (6%)	0.523 ^1^	
Psychological well-being (Score 0–25)		Total	13.4 (4.6)	13.7 (4.5)	0.263 ^3^	

^1^ McNemar Test, ^2^ Wilcoxon Test, ^3^
*t* Test.

**Table 4 healthcare-10-00002-t004:** Changes in health literacy over time in the different branches.

Branch	HL T0 $\bar{x}$ **(SD)**	HL T1 $\bar{x}$ **(SD)**	Δ HL T1–T0
Office	11.9 (3.2)	11.9 (3.1)	0.06
Retail	10.9 (3.4)	10.9 (3.3)	0.04
Education	12.0 (2.8)	12.3 (2.4)	0.30
Nursing/Medical assistant	12.1 (2.8)	12.5 (2.9)	0.47 *
Engineering	12.4 (2.8)	12.4 (3.2)	0.01
Hairdressing	11.8 (3.8)	12.6 (3.3)	0.75

* *p* = 0.019.

**Table 5 healthcare-10-00002-t005:** Multivariate logistic regression: psychological well-being.

Outcome: Psychological Well-Being (*n* = 365)		
Outcome: Lower Psychological Well-Being (Score < 37%)		
Missing Data: 6%, r2: 8%, Hosmer-Lemeshow Goodness-of-Fit Test: *p* = 0.771		
	OR * (95% CI)	*p*
HL adequate	1	-
HL problematic	2.1 (1.30–3.38)	0.002
HL inadequate	3.3 (1.70–6.32)	<0.001
Gender: female vs. male	1.9 (1.06–3.35)	0.032

* adjusted for age.

**Table 6 healthcare-10-00002-t006:** Multivariate logistic regression: subjective health status.

Outcome: Subjective Health Status (*n* = 362)		
Outcome: Poor/Less Good (15%)		
Missing Data: 7%, r2: 9%, Hosmer-Lemeshow-Goodness-of-Fit Test: *p* = 0.923		
	OR * (95% CI)	*p*
HL adequate	1	-
HL problematic	1.6 (0.84–3.22)	0.147
HL inadequate	2.8 (1.23–6.33)	0.014
Health behaviour good	1	-
Health behaviour moderate	2.2 (1.11–4.44)	0.023
Health behaviour poor	4.0 (1.51–10.48)	0.005

* adjusted for age and gender.

## Data Availability

The data presented in this study are available on request from the corresponding author.

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
