# Peer review of "How Does Health Literacy Modify Indicators of Health Behaviour and of Health? A Longitudinal Study with Trainees in North Germany"

_healthcare, 2021, doi:10.3390/healthcare10010002_

Round 1

Reviewer 1 Report

Overall, the authors presented the relationship between health literacy and health behavior in a selected group of subjects in vocational colleges in North Germany.

In Tables 5 and 6, the authors can report the goodness-of-fit test p-values and AUCs.

It appears to better if the authors include both the relationship between HL at T0 and HB at T1 as well as HL at T1 and HB at T1 in Figure 2.

Discussion

It is interesting to check whether or not the average HL scores in Table 4 are higher than those in other studies. If the average scores are already high enough, it is not easy to increase the scores.  Also, participants’ dropout is an issue. The authors can provide the comparison of HL, HS, and HB measures between the participants at T0 and T1  and the participants at T0 only.

Author Response

How does health literacy modify indicators of health behaviour and of health? A longitudinal study with trainees in North Germany

#

Reviewer 1

Author

1

In Tables 5 and 6, the authors can report the goodness-of-fit test p-values and AUCs.

Thank you for this suggestion. We added now the p-values of the Homer-Lemeshow- goodness-of-fit test to the tables.

2

It appears to better if the authors include both the relationship between HL at T0 and HB at T1 as well as HL at T1 and HB at T1 in Figure 2.

Thank you for this idea. Now we added the boxplots of HL at T1 into the diagram.

3

It is interesting to check whether or not the average HL scores in Table 4 are higher than those in other studies. If the average scores are already high enough, it is not easy to increase the scores.  

Thank you for this suggestion. That is a good thought. In relation to the scale range 0-16, the observed mean value of 12 is 75% of the scale. However, HL estimates from comparable studies are almost always available as categories in relative frequencies, so that a comparison with several studies is only possible at this level. In order to better elaborate this point, we have included further comparative studies in the discussion section:

„The prevalence of limited HL has also been reported for younger subjects, namely 58% in 15-year old Austrians and 69% students of a German college of health (28, 30). It is possible that the students from the college of health sector answered a little more critically from their own perspective, so that the proportion of limited HL is higher in this case than in comparison samples from this age group. In a survey conducted in Germany in 2014, Berens et al. found a similarly high prevalence of limited HL (47%) for the age group 15-29 years as in our study (13). In another German study performed in 2016, an equally high prevalence of 47% was observed among students aged 20-29 (37). In summary, the observed prevalence of limited HL appears to be in line with existing study findings.“

4

Also, participants’ dropout is an issue. The authors can provide the comparison of HL, HS, and HB measures between the participants at T0 and T1  and the participants at T0 only.

Thank you for this important point about assessing the potential for selection bias. To do this, we looked again at the socio-gemographic data as well as HL and health-related data for both samples. The following was added to the limitations section:

„Another factor predicting dropout was male gender, with a lower proportion of males in the follow-up than at baseline (21% vs. 30%). Regarding the health indicators (BMI, psychological well-being, subjective health status), there were no differences between the baseline sample and the follow-up sample, for the health behaviour (diet, smoking, alcohol consumption, exercise), the follow-up sample showed a lower proportion of smokers than the baseline sample at T0 (30% vs. 42%).“

5

Dear reviewer,

thank you very much for taking so much time and attention to review our manuscript in order o improve its quality.

Reviewer 2 Report

Dear Authors,

You are dealing with a relevant issue against the backdrop of the ageing of the labour force and the increasing working lifetime.

In the following, please find my comments on your submission.

Abstract: too long, contains too much details and contains an abbrevation (HHL) that is not explained. Conclusions too fuzzy.

Keywords: reconsider some of the keywords (for instance “career-starters”).

Introduction: needs more content-related structure, which could be supported by the insertion of sub-headings.

Findings for Germany lack (international) context.

At present, this chapter consists of a string of information, textual bridging is missing.

Discussion chapter contains relevant information for the Introduction chapter. Please, shift it to this section.

The insertion of a paragraph on health measurement instruments would be fine. Moreover, the provision of information on the state of play of taking into account of health literacy in the different curricula.

In this chapter policy relevant and policy prescriptive aspects tendencially are mixed up.

Sub-heading 1.1 is not necessary.

Health literacy à HHL?? (p.1)

Direct quotation on page 2: page number is missing

  1. 3, first sentence: shift to the Discussion section.

Insert a sub-heading that refers to the knowledge gap and the aim of the paper.

The hypotheses lack context to the “branches” – in my opinion “professional or rather vocational sector” would be the more accurate term.

Materials and Methods:

This section should present the research design of the “longitudinal-study” and thus should contain information on the baseline study, the methodology of the follow-up study, including the data collection process, the selection of the “survey instruments”, the contents of the questionnaire (HLS-EU-Q16) as well as the “classification” “good-intermediate-poor”.

Please explain the methodology of dichotomisation (p.4.), for instance: sporting activity, 2-4 hours /week; 1-2 hours/week (class thresholds!!)

Moreover, point out that only about 12 % of the 5052 trainees took part in the follow-up study.

Results: insert an introductory paragraph. Provide more content-related structure in order to increase readability. Insert sub-headings, for instance: description of the sample / demographic profile of respondents etc.

The placement of illustrations within the text needs to be reconsidered. (This applies for most of the tables as well as for the figures.) E.g. insert table 4 on page 7 above sub-heading 3.2.

The paragraph below Table 1 is difficult to read and confusing, e.g. “70 % of trainees in retail trade were women, in comparison with 63 % in office work.” So what?

Name of sub-heading 3.3 (p.8) should be reconsidered.

Last paragraph on page 9 should be shifted to discussion.

Please, refer more to the different “branches”.

Discussion:

Insert more sources in the first paragraph of this chapter and provide more information on the results of Jordan & Hoebel (2015).

Paragraph 4.1 contains contradictory statements. Moreover, please discuss the comparability of the study you are referring to (Schaeffer et al.) (p.10).

Paragraph 4.2. is not sufficiently supported by the results or rather findings of your longitudinal study. Insert source on the issue of “plausibility” of the positive effects of professional training related to health education than of other sectors.

Paragraph 4.3 contains information that should be shifted to the Introduction section. This applies in a large extent to the first part as well as to the whole second part of the paragraph. The same applies for parts of paragraph 4.4.

Moreover, please, ecplain the relevance of the findings of Fiedler et al as well as of Amoah et al. for the subject of this paper.

Paragraph 4.4: Insert sources in the issue of unhealthy lifestyles due to the lack of “first-hand experience of illness” (p.12).

Paragraph 4.5: shift second part of this paragraph to the Conclusion section.

Paragraph 4.6: please, reflect the following statement more critically: “Thus, our research provides consistent results that are qualitatively better.” Bear in mind the small sample size according to trainees that participated in both surveys. I fail to see the points of the statements of the last three sentences of this paragraphs.

Conclusions:

Please, provide conclusions and implications for policy and practice (incl. the development of the curricula).

Informed consent statement: Did the participants provide oral or written consent?

All the best.

Author Response

How does health literacy modify indicators of health behaviour and of health? A longitudinal study with trainees in North Germany

#

Reviewer 2

Author

1

Abstract: too long, contains too much details and contains an abbrevation (HHL) that is not explained. Conclusions too fuzzy.

Thank you for this hint. We corrected the abbriviation and shortened the abstract.

2

Keywords: reconsider some of the keywords (for instance “career-starters”).

Thank you. We changed „career-starters“ into „trainees“.

3

Introduction: needs more content-related structure, which could be supported by the insertion of sub-headings.

Thank you for your recommendation. We added three more sub-headings to the introduction to point out the structure of the text:

1.1. Definition of HL

1.2. Prevalence of limited HL, determinants and associated outcomes

1.3. Target group

4

Findings for Germany lack (international) context.

This is a good point. We cited data of a European comparison in health literacy:

„For the European region, the European Health Literacy Survey performed in 2011 found on average about one in two people has limited HL (7). Of the eight European countries studied, study participants from Bulgaria had the highest proportion of limited HL (62%), and the second highest proportion was observed among study participants from Austria (56%). The lowest proportion of impaired HL was observed in the Netherlands (29%) and Ireland (40%). With a share of 46%, the German participants were in the middle of the European comparison. In more recent studies from Germany , more than half the population (54 %) exhibit impaired HL (8).

5

At present, this chapter consists of a string of information, textual bridging is missing.

Thank you. We think that your suggestion on the sub-headings and the wording of the research gap (next point) is addressing this comment.

6

Discussion chapter contains relevant information for the Introduction chapter. Please, shift it to this section.

We added two points in the end of the introduction in order to highlight the research gap:

„Due to the small number of studies on HL and health outcomes in young adolescents and the inconsistency of these study results (22), we would like to provide further evidence to this research topic. This also concerns the inconsistent study results on HL and health behaviour in adolescents, which are in contrast to the consistent study results in adults (23).“

7

The insertion of a paragraph on health measurement instruments would be fine. Moreover, the provision of information on the state of play of taking into account of health literacy in the different curricula.

Thank you for this suggestion. The cited reviews dealing with HL and health indicators summarize studies that assessed health with many different instruments. We think the presentation of these instruments would go beyond the scope of this section.

We added this information (HL in curricula) later in the discussion section to explain our findings:

„When looking through the framework curricula of the different training professions in the six branches, we discovered that the plan for nursing professions is the only one that deals with one’s one approach of health.”

8

In this chapter policy relevant and policy prescriptive aspects tendencially are mixed up.

Sorry I cannot interpret your comment, could you please be more specific? Thank you.

9

Sub-heading 1.1 is not necessary.

We changed the sub-heading into „Research Question“.

10

Health literacy à HHL?? (p.1)

We corrected the abbriviation into „HL“.

11

Direct quotation on page 2: page number is missing

There was a back translation mistake in the definition oh HL, thank you for this hint. The defintion in original wording is:

„Health literacy is linked to literacy and entails people’s

knowledge, motivation and competences to

access, understand, appraise, and apply health information

in order to make judgments and take decisions

in everyday life concerning healthcare, disease

prevention and health promotion to maintain or

improve quality of life during the life course.“

The text has been corrected and the page number has been added.

12

 p. 3, first sentence: shift to the Discussion section.

Thank you for this suggestion. This sentence („As there has been little research on HL in occupational trainees, we regard this study as an essential contribution to research on health support and prevention in adolescents in Germany.“)  describes the motivation to perform this study in the research field. In our opinion it belongs to the introduction section.

13

 Insert a sub-heading that refers to the knowledge gap and the aim of the paper.

Under the sub-heading „Research question“ the reserach gap has been pointed out now.

14

 The hypotheses lack context to the “branches” – in my opinion “professional or rather vocational sector” would be the more accurate term.

Thank you for this point. This is right. We declared the description of HL, health and health behaviour in the subgroups over time to be explorative, for hypothesis generation so to speak. For that purpose we avoided using a hypothesis. We amended the last point into a research question:

„For hypothesis generation, we formulated the following research question: How are the changes of HL, health behaviour and health in the different subgroups (branches, genders) over time?“

15

 Materials and Methods:

This section should present the research design of the “longitudinal-study” and thus should contain information on the baseline study, the methodology of the follow-up study, including the data collection process, the selection of the “survey instruments”, the contents of the questionnaire (HLS-EU-Q16) as well as the “classification” “good-intermediate-poor”.

Thank you for this point. In the beginning of the methods section the study design has now been specified:

„The study has a prospective cohort study design.“

Information about baseline study, data collection and the different instruments are given in the text.

The classification of HL has been referred to literature:

„According to literature, HL was classified into 3 levels: adequate (13-16 points), problematic (9-12 points) and inadequate (0-8 points). If values were missing for more than two items, the total score was rated as missing.”

16

 Please explain the methodology of dichotomisation (p.4.), for instance: sporting activity, 2-4 hours /week; 1-2 hours/week (class thresholds!!)

Thank you for finding the mistake in the class thresholds, the labels have been corrected now.

„The frequency of sporting activity was surveyed on the basis of five categories (none/<1h per week/1-<2h per week/2-<4h per week/>= 4h per week) (33)”.

The dichotomisation of sporting activity followed the methodology in cited literature (see above).

17

 Moreover, point out that only about 12 % of the 5052 trainees took part in the follow-up study.

Thank you for this recommendation. To characterize the response behaviour, we have reported the response rate of the vocational colleges (14.6%), the response rate of trainess at baseline (35.5%) and the follow up rate of the follow up (27%) in the introduction section.

In our opinion, it is misleading to present the quotient of the participants of the follow-up and the persons contacted in the baseline study, since the non-responders of the baseline study had no chance to participate in the follow-up study.

18

 Results: insert an introductory paragraph. Provide more content-related structure in order to increase readability. Insert sub-headings, for instance: description of the sample / demographic profile of respondents etc.

The placement of illustrations within the text needs to be reconsidered. (This applies for most of the tables as well as for the figures.) E.g. insert table 4 on page 7 above sub-heading 3.2.

Thank you for this hint. For the first part of the result section we inserted the sub-heading: „Description of the study cohort“, all in all there are now four sub-headinds in the results section. The placement of tables and figures has been amended. I think the way we put the objects has amended now the readability of this section.

19

The paragraph below Table 1 is difficult to read and confusing, e.g. “70 % of trainees in retail trade were women, in comparison with 63 % in office work.” So what?

Thank you for this suggestion, by deletion of this sentence we amended the text as follows:

„The highest proportion of women was in the group nursing / medical assistants (95%). Most trainees came from the vocational training schools in Lower Saxony (76%) or in Schleswig-Holstein (20%).“

20

Name of sub-heading 3.3 (p.8) should be reconsidered.

Thank you, good point. We changed the name of this sub-section into: Associations of health and health behaviour with HL

21

Last paragraph on page 9 should be shifted to discussion.

Although no detailed results are presented here, information on the associations between HL and BMI or health behaviour is given here. Even if no associations or implausible associations were found, we think, that these results formally belong in the results section. Information on data not mentioned in the results section cannot, in our opinion, be addressed in the discussion section. So we would rather leave this paragragh in the results section.

22

Please, refer more to the different “branches”.

If by this comment you mean that the sectors should be taken into account in the regression models, this was not easy to realise due to small case numbers. This concerns stratification and the consideration as a covariate in the model.

23

Insert more sources in the first paragraph of this chapter and provide more information on the results of Jordan & Hoebel (2015).

Thank you. As the study of Jordan & Hoebel delivers results of HL of German adult population, we would rather presented further results of more comparible samples, of adolescents:

„It is possible that the students from the college of health sector answered a little more critically from their own perspective, so that the proportion of limited HL is higher in this case than in comparison samples from this age group. In a survey conducted in Germany in 2014, Berens et al. found a similarly high prevalence of limited HL (47%) for the age group 15-29 years as in our study (13). In another German study performed in 2016, an equally high prevalence of 47% was observed among students aged 20-29 (37). In summary, the observed prevalence of limited HL appears to be in line with existing study findings.“

24

Paragraph 4.1 contains contradictory statements. Moreover, please discuss the comparability of the study you are referring to (Schaeffer et al.) (p.10).

Thank you very much for pointing this out. We have tried to invalidate the competing explanations, but would like to retain both explanations, as we cannot rule out either of them with certainty:

„This discrepancy can certainly be explained by the narrow age range in the present study. This finding is probably unsystematic. On the other hand, one cannot exclude that, within this age range, the trainees acquire experience in health and the health system with age.  ….………..

It must also be said that the comparison of an adult sample with our data is of course limited.“

25

Paragraph 4.2. is not sufficiently supported by the results or rather findings of your longitudinal study. Insert source on the issue of “plausibility” of the positive effects of professional training related to health education than of other sectors.

It is good that we can emphasise the following again at this point: This paragraph discusses the results on the research question that was posed for hypothesis generation. Unlike the first two study questions, this is not about hypothesis testing. It was intended to answer the open question of whether the development of HL, health and health behaviour is different in one of the subgroups (gender or branch).

So we added the following:

„When looking through the framework curricula of the different training professions in the six branches, we discovered that the plan for nursing professions is the only one that deals with one’s one approach of health…………..

Therefore, we cannot exclude that the observed increase in HL for the group nursing / medical assistants is a consequence of dealing with the health of others and one's own health in the setting of the vocational school and the company. Appropriate studies need to be carried out to verify this indication.“

26

Paragraph 4.3 contains information that should be shifted to the Introduction section. This applies in a large extent to the first part as well as to the whole second part of the paragraph. The same applies for parts of paragraph 4.4.

Thank you. In our opinion, in the discussion section the study results should be compared with the existing results of literature. Of course, existing research results are also cited here, as is the case in the introduction section. We added the following to the text:

„In summary, our findings confirm the few existing studies on the relationship between HL and psychological well-being in adolescents.“

AND:

„With regard to the relationship between HL and subjective health, our study adds positive results to the few existing and controversial studies. Furthermore, our study provides results for both questions regarding psychological well-being and subjective health based on longitudinal data.“

27

Moreover, please, ecplain the relevance of the findings of Fiedler et al as well as of Amoah et al. for the subject of this paper.

Thank you for this suggestion. Due to the very limited number of studies on the relationship between HL and psychological well-being in adolescents, we have also cited the existing studies in adults, which are not optimal comparative studies due to their age.

28

Paragraph 4.4: Insert sources in the issue of unhealthy lifestyles due to the lack of “first-hand experience of illness” (p.12).

We deleted the following sentence:

„It is also possible that young trainees with high HL behave unhealthily – simply because they have had no first-hand experience of illness.“

29

Paragraph 4.5: shift second part of this paragraph to the Conclusion section.

This is a good point. We left this part in the section but we added the following in the conclusions section:

„For these at-risk groups, school and workplace-based intervention programmes that also increase HL should be implemented.“

30

Paragraph 4.6: please, reflect the following statement more critically: “Thus, our research provides consistent results that are qualitatively better.” Bear in mind the small sample size according to trainees that participated in both surveys. I fail to see the points of the statements of the last three sentences of this paragraphs.

Thank you for this suggestion. We deleted this sentence.

31

Conclusions:

Please, provide conclusions and implications for policy and practice (incl. the development of the curricula).

Thank you for this suggestion. We added the following two sentences:

„However, this exploratory result should be used in further appropriate studies to examine the extent to which curricula of health-related occupations actually increase HL over time compared to other curricula.”

AND

„For these at-risk groups, school and workplace-based intervention programmes that also increase HL should be implemented.“

32

Informed consent statement: Did the participants provide oral or written consent?

We amemded the informed consent statement into the following: „Written informed consent was obtained from all subjects invoved in the study“.

Dear reviewer,

thank you very much for taking so much time and attention to review our manuscript in order o improve its quality.

Round 2

Reviewer 2 Report

Dear Authors,

I enjoyed reading the revised version of your manuscript.

All the best.